# NIS3D: A Completely Annotated Benchmark for Dense 3D Nuclei Image Segmentation

**Wei Zheng[1], James Cheng Peng[1], Zeyuan Hou[1],**
**Boyu Lyu[1], Mengfan Wang[1], Xuelong Mi[1]**
**Shuoxuan Qiao[1], Yinan Wan[2], Guoqiang Yu[1]**
[1]Dept. of Electrical and Computer Engineering, Virginia Tech
[2]Biozentrum, University of Basel
[1]{zhengw,jameschengpeng,stanhou,boyu93,mengfanw,mixl18,kevinqiao,yug}@vt.edu
[2]{yinan.wan}@unibas.ch

## Abstract

3D segmentation of nuclei images is a fundamental task for many biological studies. Despite the rapid advances of large-volume 3D imaging acquisition methods and the emergence of sophisticated algorithms to segment the nuclei in recent years, a benchmark with all cells completely annotated is still missing, making it hard to accurately assess and further improve the performance of the algorithms. The existing nuclei segmentation benchmarks either worked on 2D only or annotated a small number of 3D cells, perhaps due to the high cost of 3D annotation for large-scale data. To fulfill the critical need, we constructed NIS3D, a 3D, high cell density, large-volume, and completely annotated Nuclei Image Segmentation benchmark, assisted by our newly designed semi-automatic annotation software. NIS3D provides more than 22,000 cells across multiple most-used species in this area. Each cell is labeled by three independent annotators, so we can measure the variability of each annotation. A confidence score is computed for each cell, allowing more nuanced testing and performance comparison. A comprehensive review on the methods of segmenting 3D dense nuclei was conducted. The benchmark was used to evaluate the performance of several selected state-of-the-art segmentation algorithms. The best of current methods is still far away from human-level accuracy, corroborating the necessity of generating such a benchmark. The testing results also demonstrated the strength and weakness of each method and pointed out the directions of further methodological development. The dataset can be downloaded here https://github.com/yu-lab-vt/NIS3D.

## 1 Introduction

With the rapid development of live-cell microscopic imaging and genetic fluorescent reporters, researchers are able to record the time-lapse 3D images of cell nuclei during the embryogenesis process[1]. Such data are valuable for a wide range of biological research, for instance, the mechanisms and patterns of cell differentiation, the origin and diversity of cell types, and the causes and consequences of developmental defects [2–7]. In these studies, a critical step is 3D embryonic cell nuclei image segmentation, which is the foundation of subsequent analyses including cell tracking, lineaging analysis, morphogenesis analysis, and morphodynamic analysis [8–10].

Submitted to the 37th Conference on Neural Information Processing Systems (NeurIPS 2023) Track on Datasets and Benchmarks. Do not distribute.

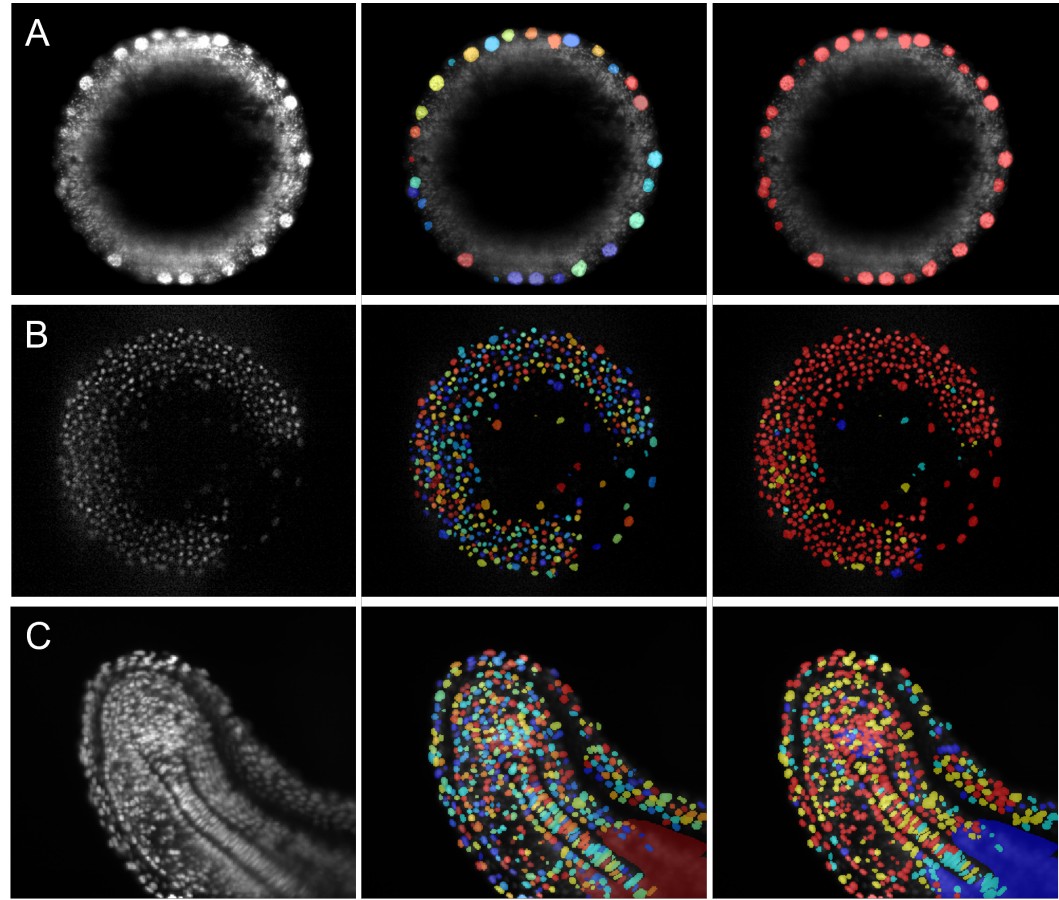

Figure 1: Examples of NIS3D benchmark. Each row represents a specific data, with columns from left to right displaying raw data, the annotated ground truth, and the corresponding confidence score map, respectively. Confidence scores are assigned on a four-level scale to indicate the reliability of each annotation, from low to high: "undefined masks", "1/3", "2/3", and "3/3". These levels are labeled by the colors deep blue, light blue, yellow, and red, respectively. It's worth noting that "undefined mask" indicates regions with a group of very blurry cells that annotators can't decide their boundaries. The detection whose majority of pixels are within the undefined masks will be ignored, neither considered as true positive nor false positive.

Unlike other types of nuclei image data, 3D embryonic nuclei data possess distinctive characteristics, such as high cell density, large volumes, low signal-to-noise ratio (SNR), and a diverse range of shapes and intensities within the same volume, as exemplified in Figure 2. Consequently, the segmentation of 3D embryonic nuclei images presents great challenges. Despite the numerous 3D segmentation methods proposed, there is currently a lack of a widely accepted comprehensive benchmark for evaluating their performance. The existing benchmarks or datasets [11–16] for nuclei segmentation predominantly provide 2D ground truth, thereby overlooking the critical aspect of 3D analysis. Although some datasets [17, 18] do offer 3D annotations, they focus on the very early stages of embryo development, resulting in uncharacteristically low cell density and a very limited number of annotated cells.

Annotating large volumes of 3D embryonic nuclei is a time-consuming and labor-intensive task that requires a thorough manual inspection of the data. Unlike 2D images, where objects reside on individual planes with relatively simple morphological structures, annotating 3D images presents significantly greater challenges for the following reasons: (a) In 3D image annotation, a nucleus is captured across multiple consecutive z-slices, resulting in 2D boundaries on each slice comprising

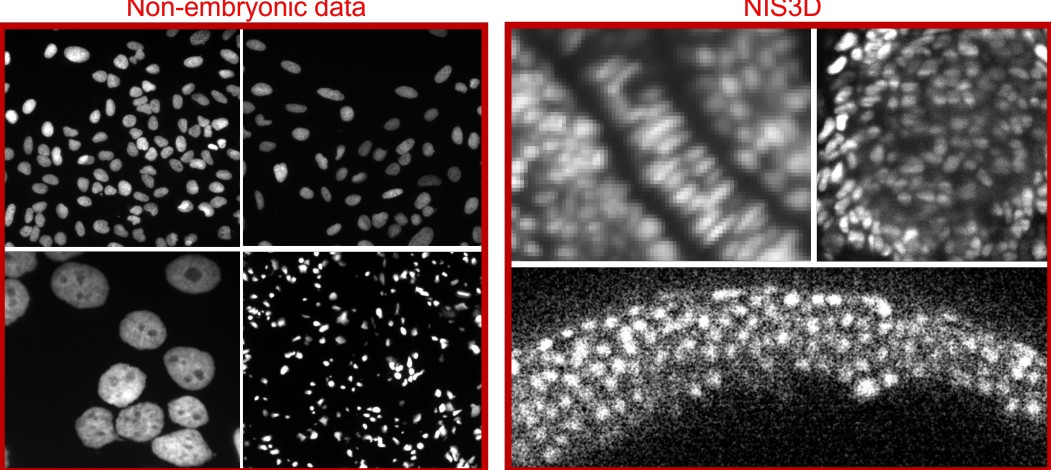

Figure 2: Left: 2D non-embryonic nuclei images [12, 13, 19]. Right: 3D embryonic nuclei images.

its surface. Consequently, annotating the same number of cells in 3D requires significantly more time compared to 2D labeling. (b) In dense object arrangements, portions of the cell surface can be parallel to the chosen visualization view and remain invisible, further complicating the annotation process. (c) The cell morphology and texture in 3D images exhibit far greater complexity than their 2D counterparts, demanding annotators to adhere to higher standards when labeling nuclei accurately. Consequently, annotating 3D images necessitates a larger investment of human labor compared to 2D cases.

To fill the gap of a completely annotated 3D embryonic cell image dataset, in this report, we present NIS3D, a 3D, high cell density, large-volume, and completely annotated embryonic Nuclei Image Segmentation benchmark. We provide examples of the benchmark in Figure 1. NIS3D provides more than 22,000 3D nuclei in the embryo images of zebrafish, drosophila, and mouse, which are the most commonly used species in the field. Each image of NIS3D is annotated by three independent well-trained annotators spending a total of 700+ hours. To allow more nuanced testing and performance comparison, a confidence score is computed for each cell to show its reliability. There are four levels of confidence scores in NIS3D, from the least confident score representing ambiguous annotation to the largest confident score indicating great consistency among all annotators.

To be more specific, the advantages of NIS3D are as follows:

- A good representation: NIS3D provides large-volume images of high cell densities, with nuclei whose signal-to-noise ratio, shape, and brightness vary with position. These properties make this benchmark challenging but well representative of data from real research.

- 3D complete annotation: All cells are annotated, and all labels are 3D. Compared with sparse annotation or 2D annotation, NIS3D can provide a more comprehensive evaluation, including the evaluation of false positives.

- Confidence score: A confidence score is computed for each cell, allowing more nuanced testing and performance comparison.

- Multiple species: NIS3D contains the three most commonly used species in this field (zebrafish, drosophila, and Mus Musculus) to provide enough diversity.

To facilitate the 3D annotation, we developed a semi-automatic annotation tool. It can generate a suggestive 3D boundary on all z-slices for the user-identified cell, without tedious labeling on each z-slice. The suggestive boundary can even outperform human annotation in low-quality regions, but annotators still have the authority and flexibility to further fine-tune unsatisfied cell boundaries. The tool not only significantly speeds up the annotation workflow but also reduces human bias.

Table 1: Existing benchmarks

| Name | Data | Label | Annotator # | Complete label | Sample # | Year |
|---|---|---|---|---|---|---|
| Cell tracking challenge[14] | 3D | 2D | 3 | No | 3,000+ | 2014 |
| BBBC039v1 [20] | 2D | 2D | 1 | Yes | 23,000+ | 2018 |
| 2018 Data Science Bowl[12] | 2D | 2D | 1 | Yes | 30,000+ | 2018 |
| S-BSST265[19] | 2D | 2D | N/A | Yes | 7813 | 2020 |
| BBBC032v1[17] | 3D | 3D | N/A | Yes | 57 | 2018 |
| BBBC050[18] | 3D | 3D | N/A | Yes | 1,814 | 2020 |
| C.elegans[21] | 3D | 3D | 1 | Yes | 15,000+ | 2022 |
| NIS3D (ours) | 3D | 3D | 3 | Yes | 22,000+ | 2023 |

*For Cell tracking challenge, we only consider the 3D embryonic data. N/A annotator number means that the work didn't mention it.

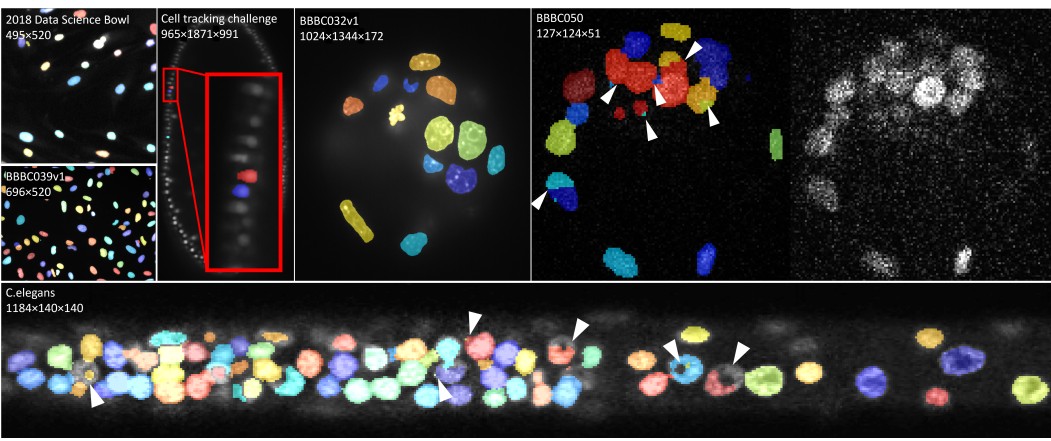

Figure 3: Representative examples of existing benchmarks. The numbers on the left top indicate the resolution of this data. All data shown here are the full data or z-slices from the corresponding dataset and are not cropped. Different colors represent different ground truth labels. The over-segment and under-segment issues in BBBC050 are indicated by white arrows.

## 2   Related work

**Existing benchmarks**   Table 1 summarizes the existing benchmarks in this field and Figure 3 shows examples of them. The cell tracking challenge is considered as the most commonly-used benchmark in this field, but it only provides sparse annotation for embryonic data, making the false positive evaluation infeasible. BBBC039v1, 2018 data science bowl, and S-BSST265 are also popular benchmarks providing nuclei data, but they are not embryonic and have a relatively low nuclei density with nuclei well separated. As a result, they cannot be used to comprehensively evaluate the segmentation algorithms dealing with densely packed nuclei. Moreover, all these benchmarks only provide 2D annotations while various biological questions require algorithms to detect the boundary in 3D space. It should be pointed out that BBBC032v1 and BBBC050 do provide 3D annotations of animal embryos of mouse and drosophila, but they are at very early developmental stages with two critical issues: unusually low nuclei density and small sample size. Moreover, BBBC050 is not initially made as a benchmark and has some quality issues. Figure 3 shows obvious under-segment and over-segment problems for the ground truth of BBBC050. *C.elegans* is a better option, but it not only suffers from low diversity but also has discrepancies such as incomplete or erroneous labels, which is hard to avoid for ground truth from only one annotator. Additional information can be found in the supplementary.

**Existing segmentation methods**   The existing segmentation methods can be classified into two groups: semantic segmentation and instance segmentation. Semantic segmentation assigns the

same label to the object of the same class. For nuclei images, it can only get foreground rather than individual cells. Some famous models, like ilastik[22], labkit[23], and 3D U-Net [24] belong to this class. Such models work well for many low cell-density data. But for embryonic data analysis, they are far from satisfactory, and thus instance segmentation is necessary. There are popular tools, such as Vaa3D[25], MorphoLibJ[26], and 3D Suite[27], providing unsupervised methods for instance segmentation. Those methods are easy to use but generally fail to deal with the complex morphological and intensity patterns of embryonic data. There are also some supervised models, for instance, methods proposed in recent years, like Mesmer[15], QCAnet[18], Cellpose[11], and StarDist [28]. Some of those methods like Mesmer can only be used for 2D images. QCAnet consists of two submodels, which can detect the foreground and nuclei center separately, then it uses the marker-based watershed to generate the segmentation result. Cellpose and StarDist are originally designed for 2D data, but they both make 3D extensions based on assumptions of 3D cell shape. The Cellpose 3D extension is still trained on 2D data, but it does 2D segmentation for xy plane, xz plane, and yz plane separately first, and then thresholds the average cell probability from 3 directions to reconstruct the 3D segmentation result. StarDist 3D extension estimates both cell probability and radial distance to the boundary for each pixel, then reconstructs the 3D segmentation result.

## 3 NIS3D

NIS3D collects 6 large volume embryonic nuclei images from the three most widely used species in the field with over 22,000+ manually annotated cells. In this section, we will give the details of data collection, annotation, and recommended evaluation metrics. Additional information can be found in the supplementary.

### 3.1 Data Collection

**Zebrafish 1 (in-house dataset)** Transgenic zebrafish embryos with fluorescent nuclei marker *Tg(bactin2:H2BmCherry)*, inside their chorions, were embedded in 1% low melting point agarose prepared in E3 medium, enclosed by glass capillary before extruded into the imaging chamber. Images were acquired with Zeiss LightSheet 7 Microscopy, with 20x/N.A. 1.0 detection objective (additional optical zoom factor 0.55x) and dual-side 10x/N.A. 0.3 illumination objectives. Fluorescence was activated by 561nm laser and detected with LP585 filter. Time-lapse imaging was performed at 2-minute interval from 4 to 20 hours post fertilization. Within each time interval, four 3D volumes were acquired with 90-degree rotation in between to achieve full-embryo multiview coverage. The z-stack was set to have the voxel size of 0.43 um x 0.43 um x 2.5 um, so that each cell nuclei is sectioned by at least 3 planes. We picked a time point in the middle of this time-lapse data. The nuclei in this data don't have strong texture and the nuclei shapes are consistent, but the data suffers from low SNR, especially in the first 40 z-slices. The voxel size is 0.43 um x 0.43 um x 2.5 um.

**Zebrafish 2** The data are the first time-point of embryo 3 in the public dataset [7]. Zebrafish 2 recorded the tailbud of a zebrafish in a late stage of embryo development. There is a significant amount of blurring within more anterior portions of the tail, which is hard even for human to distinguish the boundary and we use the undefined mask to mask it out. This data have very high cell density and small cell size. The voxel size is 1 um x 1 um x 1 um.

**Drosophila (fruit fly) 1&2** The two drosophila images are selected from [10]. Drosophila images are picked at time-point 20 and 50 of the brachyenteron (byn) gene reporter data. The data are for early Drosophila embryogenesis and the cells are large and relatively sparse. This data shows complicated textures within cells and bright background noise. The voxel size is 1 um x 1 um x 1 um.

**Mus musculus (mouse) 1&2** The Mus musculus data are selected from [9]. The image is picked at the time-point 150 and 200 of embryo 4. It was recorded during relatively late Mus musculus embryogenesis. The center of this image is very blurry, and we use an undefined mask to mask it.

This data shows various cell shapes and strong textures within cells. The voxel size is 1 um x 1 um x 1 um.

## 3.2 Data annotation

The workflow consists of three steps: independent annotation, label fusion, and manual reviewing.

**Independent annotation** We first train our annotators on how to distinguish the nuclei, noise, background, texture within cells, and the gap between cells for nuclei images. Then we train the annotator how to use PrinCut, the semi-automatic annotation tool we developed for this project. PrinCut can automatically generate a suggestive boundary for the user-identified cell at the surface with continuous positive principal curvature. The suggestive boundary is sensitive to the weak intensity changes that humans may ignore, which can reduce human bias in low-quality regions. However, principal curvature can also be over-sensitive to cell texture and insensitive to the shape of cells. We request the annotators to merge and split the suggestive boundaries or manually draw the boundary by brush until the boundaries meet the annotator's expectations. For the region that annotators do believe there are cells but cannot identify the boundaries due to image quality, annotators will label it as an undefined mask. By the end of this step, we get three sets of independent labels for each image.

**Label fusion** We first match the labels from different annotators, as shown in Figure 4. We consider labels from three annotators as matched if they have an intersection over union (IoU) greater than 0.5 between each other. For labels that meet these criteria, we consider them to be associated with the same ground truth, and we calculate the boundary of this ground truth label based on the three matched labels (more details in supplementary). The same approach is applied to find all ground truths and their corresponding labels. If the ground truth is associated with $x$ labels from different annotators, its confidence score is $x/3$. About 1.76% of labels belong to conflict labels, which usually means the regions in those labels are very confusing. Those labels will be further manually reviewed.

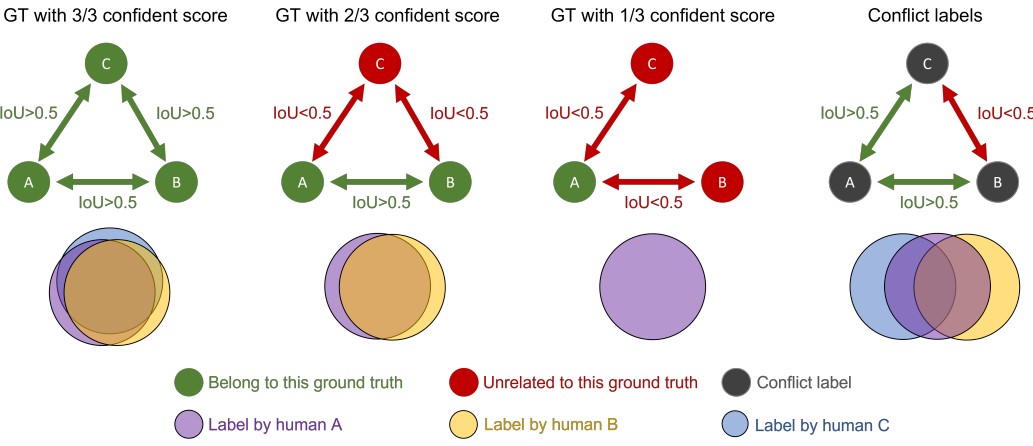

Figure 4: The top row shows criteria for determining the labels belonging to the same ground truth, while the bottom row shows an example for this case. The three circles row represent the three labels from different annotators and we will calculate the Intersection over Union (IoU) between them. "IoU<0.5" generally means that two labels belong to different ground truths. "IoU>0.5" generally means that two labels belong to the same ground truth. The green circles in the top row represent the labels belonging to the ground truth while the red circles represent the label unrelated to this ground truth. For example, annotator A and annotator B created annotation for 2/3 confidence score ground truth while annotator C didn't create any annotation for this ground truth, and the confidence score of this ground truth is 2/3.

Table 2: Confidence score distribution.

| Data \ Score | 1/3 | 2/3 | 3/3 |
|---|---|---|---|
| Zebrafish 1 | 529 | 1762 | 10423 |
| Zebrafish 2 | 1668 | 1745 | 2224 |
| Drosophila 1 | 13 | 85 | 1596 |
| Drosophila 2 | 1 | 1 | 488 |
| Mus Musculus 1 | N/A | 369 | 1191 |
| Mus Musculus 2 | N/A | 395 | 660 |

By the end of this step, we get a set of ground truth labels with different confidence scores and a group of conflict labels for each image.

**Manual reviewing** For the conflict label with bad image quality, the label will be manually set as an undefined mask, otherwise, we pick the best candidate label as the ground truth label and set the confidence score as 1/3. For low-quality data, extra undefined masks are also drawn on specific low-quality regions. The Mus Musculus images have strong textures within cells, we manually set all ground truth with 1/3 confidence score to uncertain labels. Table 2 shows the distribution of the confidence score as a reference of human annotation variation.

## 3.3 Evaluation metrics

Choosing the correct metric that adequately reflects the biological nature is important but usually neglected [29]. The existing metric of the cell tracking challenge and the 2018 Data Science Bowl give results that are inconsistent with human intuition, thereby affecting the evaluation process. To address these issues, we have reformulated the evaluation metric to align more closely with our specific objectives. For instance, a high W-F1 score coupled with a low W-SEG score now indicates successful cell detection while indicating room for boundary enhancement. Similarly, a high W-IoU score combined with a low W-SEG score signifies accurate foreground detection, while highlighting potential over-segmentation or under-segmentation concerns.

**Preprocessing and truth positive criteria** To initiate the process, we exclude detections where more than 50% of their pixels fall within the undefined mask. Then for a given detection $D_i$, we determine it matches with ground truth $G_j$ if and only if both of the two following condition holds:

$$D_i = \arg\max_{D_k} \text{IoU}(D_k, G_j) \qquad G_j = \arg\max_{G_k} \text{IoU}(D_i, G_k), \qquad (1)$$

where $D_k$ and $G_k$ are all possible choices from detections and ground truth, and $\text{IoU}(A, B)$ is the intersection over the union between $A$ and $B$.

**Weighted precision(W-Precision), recall(W-Recall), and F1(W-F1)** The weighted scores are based on the confidence score.

The weighted true positive (W-TP) and false negative (W-FN) are calculated as follows:

$$\text{W-TP} = \sum C_i T_i \qquad \text{W-FN} = \sum C_i(1 - T_i), \qquad (2)$$

where $C_i$ is the confidence score of ground truth $G_i$ and $T_i$ is the detection flag of $G_i$. $T_i = 1$ indicates if $G_i$ is detected, otherwise $T_i = 0$.

The weighted precision (W-Precision), recall (W-Recall), and F1 (W-F1) are calculated as follows:

$$\text{W-Precision} = \frac{\text{W-TP}}{\text{W-TP} + \text{FP}} \quad \text{W-Recall} = \frac{\text{W-TP}}{\text{W-TP} + \text{W-FN}} \quad \text{W-F1} = \frac{2\text{W-TP}}{2\text{W-TP} + \text{FP} + \text{W-FN}} \quad (3)$$

**Weighted IoU** Weighted IoU (W-IoU) is used to show the accuracy of foreground of detection.

$$\text{W-IoU} = \frac{\sum_{i \in A \cap B} f(i)}{\sum_{i \in A} f(i) + \sum_{i \in B/A} 1} \qquad (4)$$

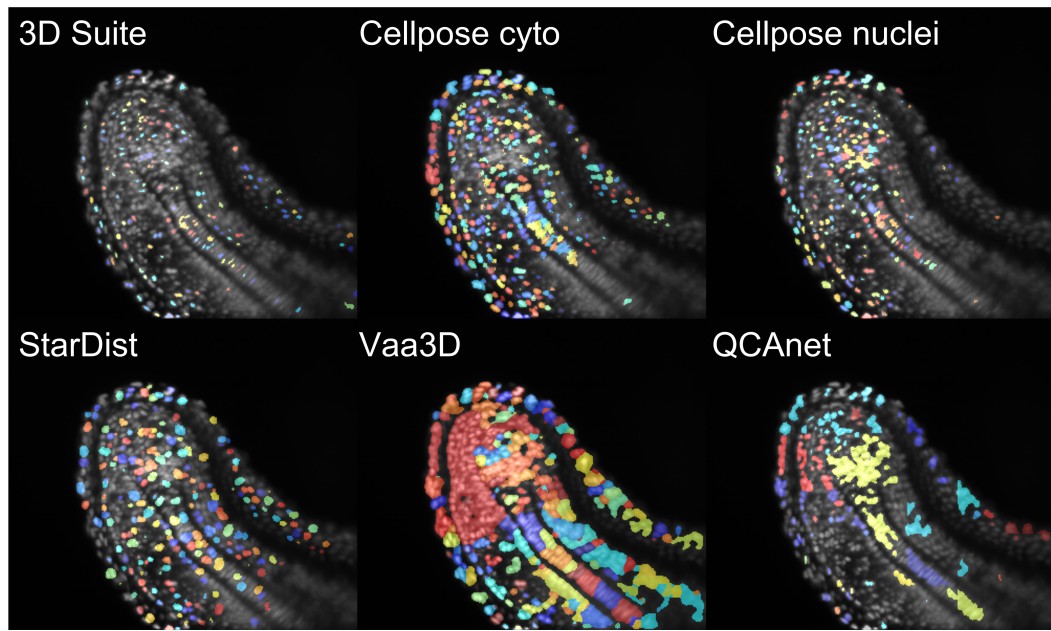

Figure 5: Examples of baseline methods results. The raw data and ground truth are from Zebrafish 2 and are also shown in Figure 1(C). The different colors represent different detection instances.

where $A$ is the pixel belong to ground truth, $B$ is the pixel belong to detection, and $f(i)$ is the confidence score of $i$-th pixel.

**Weighted SEG**   Weighted SEG (W-SEG) score is used to show the average IoU score of all cells.

$$\text{W-SEG} = \frac{\sum C_i G_i}{\text{W-TP} + \text{W-FN} + \text{FP}}, \tag{5}$$

where $G_i$ is the IoU score of the $i$-th ground truth.

## 4   Experiments

In this section, we test baseline methods on our benchmark dataset. The baseline methods we choose are: 3D Suite [27], Cellpose [11], StarDist [28], Vaa3D [25], and QCAnet [18]. The primary reason for selecting these particular methods over others is their capability to perform 3D instance segmentation and their relatively better performance, combined with the fact that the developers have provided software with a user-friendly interface accompanied by detailed instructions for usage. In this way, it is feasible for us to tune the methods on the data in NIS3D so that we can provide a fair performance analysis. In this section, we will present the experimental settings, evaluate the results, and discuss the limitation of baseline methods.

### 4.1   Experimental settings

All experiments are conducted on our workstation with NVIDIA V100 GPU and Intel Xeon Platinum 8268 CPU in this work. We provide the correct cell size and resolution as parameters for all methods. 3D suite includes multiple unsupervised methods, where we choose iterative thresholding due to its best performance. Cellpose also provides several models, and we choose "cyto" and "nuclei" for the same reason. In this section, the results of supervised methods are based on pre-trained models because some of them can only be trained on 2D data and it is unfair to train others. Other settings for each method are summarized in the supplementary.

## 4.2 Benchmark result

The experiment results of baseline methods are summarized in Table 3 and examples of the results are visualized in Figure 5. Evaluation metrics are introduced in Section 3.3. The IoU represents the intersection over the union between foreground and background to evaluate the binary segmentation performance, while SEG represents the average IoU treating every cell individually to evaluate the instance segmentation performance. When a method's SEG score is much lower than the IoU score, it generally means that this method has a lot of instance segmentation errors, such as under-segment multiple cells or over-segment one cell. 3D suite, an iterative threshold method, tends to only detect the bright region of cells and miss many dim cells. As a result, it usually gives high precision but low recall and SEG. The performance of Cellpose cyto varies from data, which can miss the majority of cells for some data and give a very low IoU score. Cellpose nuclei tend to over-segment

Table 3: The table of experiment result.

| Data name | Method name | W-F1 | W-Prec | W-Recall | W-IoU | W-SEG | Time (s) |
|---|---|---|---|---|---|---|---|
| Zebrafish 1 | 3D Suite | 0.477 | 0.998 | 0.313 | 0.102 | 0.076 | 718 |
| | Cellpose_cyto | 0.098 | 0.769 | 0.053 | 0.013 | 0.007 | 1271 |
| | Cellpose_nuclei | 0.492 | 0.363 | 0.761 | 0.339 | 0.073 | 1303 |
| | StarDist | 0.586 | 0.770 | 0.473 | 0.263 | 0.192 | 1620 |
| | Vaa3D | 0.041 | 0.992 | 0.021 | 0.209 | 0.002 | 1200 |
| | QCAnet | 0.510 | 0.885 | 0.359 | 0.276 | 0.158 | 8070 |
| | Human | 0.967 | 0.971 | 0.963 | 0.927 | 0.905 | N/A |
| Zebrafish 2 | 3D Suite | 0.462 | 0.985 | 0.302 | 0.110 | 0.066 | 180 |
| | Cellpose_cyto | 0.661 | 0.949 | 0.507 | 0.492 | 0.237 | 740 |
| | Cellpose_nuclei | 0.654 | 0.971 | 0.492 | 0.263 | 0.167 | 242 |
| | StarDist | 0.529 | 0.977 | 0.363 | 0.380 | 0.183 | 60 |
| | Vaa3D | 0.101 | 0.995 | 0.053 | 0.703 | 0.007 | 56 |
| | QCAnet | 0.031 | 0.849 | 0.016 | 0.446 | 0.003 | 496 |
| | Human | 0.880 | 0.915 | 0.848 | 0.805 | 0.610 | N/A |
| Drosophila 1 | 3D Suite | 0.879 | 0.859 | 0.900 | 0.265 | 0.207 | 362 |
| | Cellpose_cyto | 0.550 | 0.397 | 0.894 | 0.815 | 0.257 | 320 |
| | Cellpose_nuclei | 0.607 | 0.457 | 0.903 | 0.421 | 0.151 | 215 |
| | StarDist | 0.924 | 0.881 | 0.972 | 0.691 | 0.586 | 480 |
| | Vaa3D | 0.260 | 0.988 | 0.149 | 0.671 | 0.025 | 278 |
| | QCAnet | 0.275 | 0.585 | 0.180 | 0.595 | 0.051 | 2334 |
| | Human | 0.997 | 0.995 | 0.998 | 0.915 | 0.906 | N/A |
| Drosophila 2 | 3D Suite | 0.877 | 0.785 | 0.993 | 0.339 | 0.273 | 420 |
| | Cellpose_cyto | 0.092 | 0.048 | 0.982 | 0.606 | 0.028 | 478 |
| | Cellpose_nuclei | 0.278 | 0.162 | 0.990 | 0.413 | 0.057 | 236 |
| | StarDist | 0.684 | 0.526 | 0.979 | 0.563 | 0.325 | 245 |
| | Vaa3D | 0.446 | 0.651 | 0.339 | 0.428 | 0.094 | 307 |
| | QCAnet | 0.402 | 0.685 | 0.284 | 0.399 | 0.133 | 2790 |
| | Human | 0.996 | 0.992 | 0.999 | 0.909 | 0.903 | N/A |
| Mus Musculus 1 | 3D Suite | 0.463 | 0.824 | 0.322 | 0.175 | 0.102 | 600 |
| | Cellpose_cyto | 0.536 | 0.509 | 0.566 | 0.339 | 0.142 | 602 |
| | Cellpose_nuclei | 0.356 | 0.240 | 0.688 | 0.37 | 0.056 | 570 |
| | StarDist | 0.594 | 0.789 | 0.476 | 0.256 | 0.191 | 480 |
| | Vaa3D | 0.207 | 1.000 | 0.116 | 0.515 | 0.018 | 780 |
| | QCAnet | 0.454 | 0.726 | 0.330 | 0.451 | 0.127 | 4133 |
| | Human | 0.981 | 0.975 | 0.988 | 0.973 | 0.897 | N/A |
| Mus Musculus 2 | 3d Suite | 0.667 | 0.841 | 0.553 | 0.127 | 0.090 | 92 |
| | Cellpose_cyto | 0.371 | 0.412 | 0.338 | 0.099 | 0.045 | 188 |
| | Cellpose_nuclei | 0.378 | 0.285 | 0.561 | 0.235 | 0.054 | 264 |
| | StarDist | 0.540 | 0.549 | 0.531 | 0.311 | 0.158 | 221 |
| | Vaa3D | 0.270 | 0.967 | 0.157 | 0.510 | 0.032 | 147 |
| | QCAnet | 0.298 | 0.689 | 0.190 | 0.391 | 0.067 | 1472 |
| | Human | 0.959 | 0.967 | 0.950 | 0.887 | 0.809 | N/A |

and give very low precision. StarDist has the overall best performance since it has relatively fewer overall under-segment or over-segment issues. However, StarDist still gives a considerable amount of false positives and false negatives. Vaa3D can usually detect the foreground well but tends to under-segment cells, which leads to high precision and IoU but low recall and SEG scores. QCAnet tends to under-segment and gives low recall as well. For some data, QCAnet may detect a large region full of noise. The human-level performance is also evaluated and provided. For human performance, the SEG score of Zebrafish 2 is lower than other data because of their smaller cell size.

## 5 Conclusion & Discussion

NIS3D presents a 3D, high cell density, large-volume, and completely annotated Nuclei Image Segmentation benchmark with over 22,000+ cells from commonly studied species in the field. To the best of our knowledge, NIS3D is the first benchmark to provide a publicly available 3D nuclei image annotation of this scale and offers method developers a valuable opportunity to comprehensively evaluate their techniques, establishing an essential foundation for further method development. For supervised models, We also provide two suggestive training/test split settings, one is an in-image split setting and one is a cross-image split setting. For the in-image split, we use 50% of the image as the training set and the other 50% as the test set. For cross-image split, we use 3 full images as the training set and the rest as the test set. check the supplementary for more details about the supervised learning result and suggestive splitting.

It is worth noting that existing segmentation methods are partially limited by their reliance on natural image segmentation principles and place the primary focus on predicting cell foregrounds. Consequently, these methods often suffer from both over-segmentation and under-segmentation issues when objects are densely packed. Considering the fact that these issues are highly related to cell boundary detection and cell boundaries are relatively easier to be detected in nuclei data, we suggest that future method developers not only estimate the probability of cells for each pixel but also estimate the probability of cell boundaries.

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
