# Supplementary

# Contents

Submitted to the 37th Conference on Neural Information Processing Systems (NeurIPS 2023) Track on Datasets and Benchmarks. Do not distribute.

# 1 PrinCut

## 1.1 How to use PrinCut

The PrinCut GUI is shown in Figure 1. PrinCut is a MATLAB app, and its package is also provided in the supplementary. The app is tested and used on MATLAB 2022b.

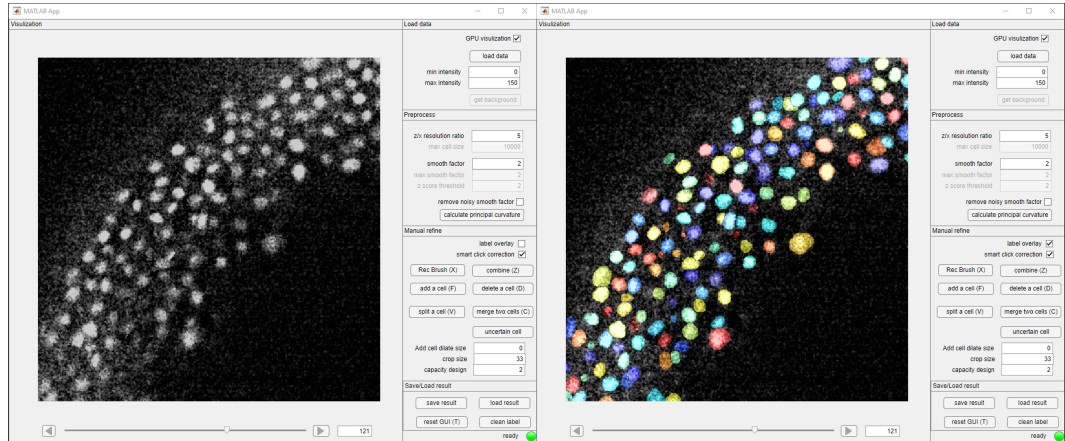

Figure 1: The GUI of PrinCut. The left shows raw data without annotation. The right shows both raw data and annotation overlay.

After loading the raw data (the data should be in tif format), users can adjust the contrast, zoom in/out, and change different z slices to visualize the data. Users then need to give the z/x resolution ratio and the smooth factor to calculate the principal curvature. For example, if the voxel size of data is $1um \times 1um \times 5um$, then the z/x resolution ratio should be 5. The smooth factor is the standard deviation (by pixel) of the Gaussian filter used to smooth data before calculating the principal curvature. After adjusting the z/x resolution ratio and the smooth factor, click the "calculate principal curvature" button.

When the principal curvature is calculated, users can add a cell label by simply clicking the cell, and a 3D suggestive boundary will be automatically generated. We request the annotators check the suggestive boundary on every z slice to make sure it's correct. If the suggestive boundary contains more than one cell, users can split the cell by clicking the center of each cell and PrinCut can give a suggestive boundary to each cell. If the suggestive boundary only contains part of a cell, users can merge two labels and PrinCut will give a suggestive boundary including the original labels. Users can also use the "combine" button to combine two existing boundaries without generating suggestive boundaries. Examples are shown in Figure 2. Four additional parameters are used to control the suggestive boundary, which are "smart click correction", "Add cell dilate size", "crop size", and "capacity design". In the methodology section, we will discuss how those parameters influence the suggestive boundary.

However, there is a chance that the gap between two cells is too weak that the principal curvature is still negative. In this case, users need to use the brush to draw the expected boundary. The size of the brush can be adjusted by users, as is shown in Figure 5.

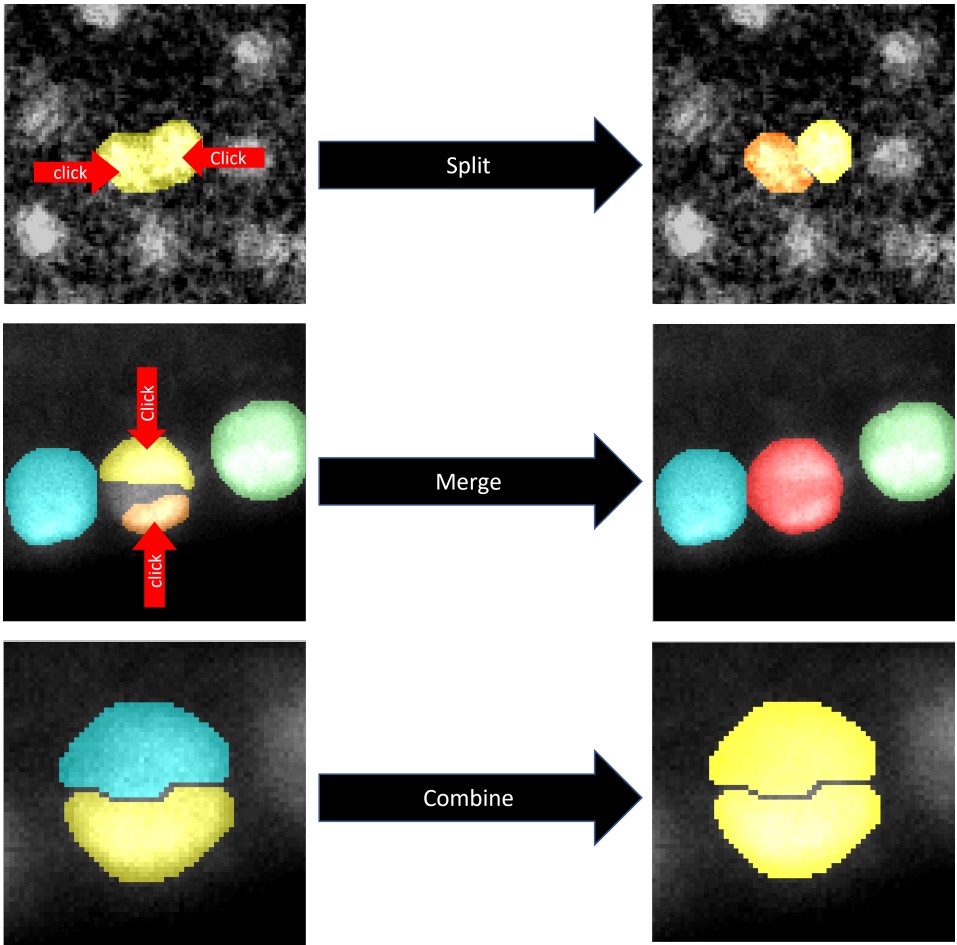

Figure 2: Examples of splitting, merging, and combining existing cells by clicking. PrinCut can automatically generate a suggestive boundary after split or merge process.

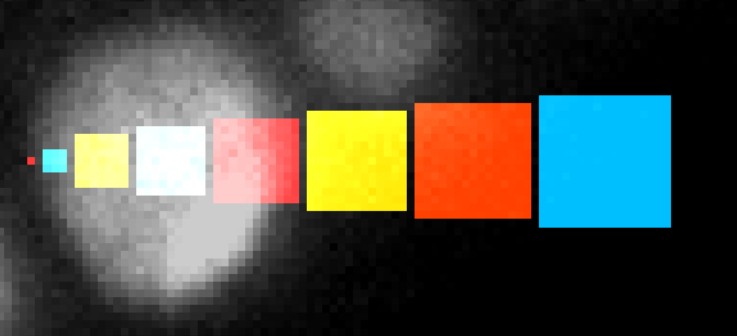

Figure 3: The brushes with different sizes, from a single pixel to $17 \times 17$ pixels.

## 1.2 Methodology

### 1.2.1 Boundary-refine algorithm and adding a cell

The boundary-refine algorithm can grow a given seed to the suggestive boundary in a given foreground. For adding a cell process, the seed is a spherical region using the pixel user clicked as the center and the "add cell dilate size" parameter in GUI as the radius, and the foreground is a spherical region with the same center and using the "crop size" parameter in GUI as the radius. The seed should be within the cell while the foreground should be larger than the cell.

The optimization criteria of the boundary-refine algorithm are to grow the seeds such that the principal curvature along the boundary pixels is maximized. However, this constraint alone is not enough, since the seed might grow to an excessively large region to meet the constraint, so we also need to add the constraint that the boundary should be as short as possible. Furthermore, since there can be multiple seeds inside one region, we need to make sure the regions grown from each seed do not interfere with each other, or the regions are not intersected with each other.

Marker-based watershed inside the foreground is one common solution to solve the problem. But it is solely based on the score map and will always grow the marker so that all the pixels are used for segmentation. But in our case, we don't want the seed to grow and fill the whole foreground, and the grown regions of all the seeds do not need to be in contact with each other. To align the boundary to the optimal position, we formulate the problem into an optimization problem.

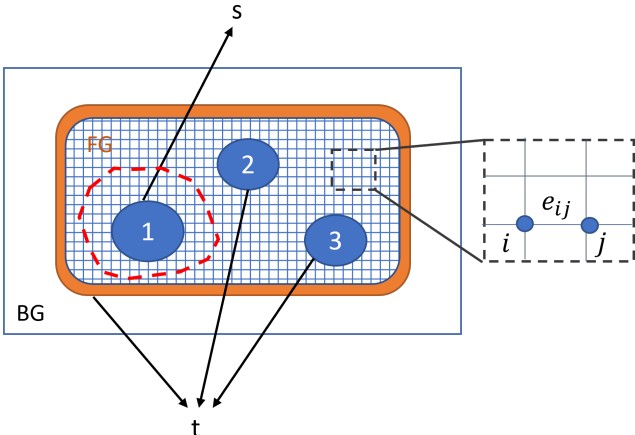

Figure 4: The graph for solving the refined boundary based on the seed

Consider the labeling of each pixel $x_i^n$ in the foreground as $u_i^n$, if $u_i^n = 1$, then $x_i$ will be assigned to the seed $n$, otherwise if $u_i^n = 0$, the corresponding pixel will be assign as outer region. And we can define the group of pixels belonging to the seed $n$ as $S_n$ and the group for the rest of pixels as $\bar{S}_n$. And the boundary between the two groups can be represented using the pairs of pixels $C_n$. In each pair, one belongs to the grown seed region, the other belongs to the background region.

$$C_n = \{(x_i, x_j) | x_i \in S_n, x_j \in \bar{S}_n\} \tag{1}$$

Then the objective function for seed $n$ can be expressed as,

$$\arg\min_{C_n} \sum_{(i,j) \in C_n} \left( \hat{G}(i) + \hat{G}(j) \right) \tag{2}$$

where $\hat{G} = \frac{1}{max(G,T)^p}$ and $G$ is the map of the principal curvature for each pixel, $p$ is the "capacity design" parameter in GUI, $T$ is a constant as 0.001. In this way, we transform the problem of finding the boundary pixels maximizing in the map $G$ into minimizing in the map $\hat{G}$. In addition, this objective function also implicitly minimizes the boundary length. In order to solve 2, we can reformulate the problem into a min-cut problem that can be efficiently solved. First, we will introduce graph construction. As shown in figure 4, the graph for optimizing based on seed 1 is built based on all the pixels in the 3D foreground, and each node $i$ represents a pixel $i$. Between each pair of neighbor nodes, an edge $e_{ij}$ is linked between them. The edge weight is defined as $weight(e_{ij}) = \hat{G}(i) + \hat{G}(j)$. In addition, one pair of pseudo nodes are added, one is the pseudo source node $s$ and the other the pseudo sink node $t$. All the pixels in the corresponding seed will be connected to the source node, and the boundary of the foreground will be connected to the sink (labeled in orange in figure 4). The weights of these two types of edges are set as infinite or very large.

With this graph design, the labeling of the nodes can induce a cut set $C = (i, j)|u(i) \neq u(j)$ and the following optimization of the labeling $U$ for seed $n$ is the same as 2

$$u^n = \underset{u_i, i \in [1,N]}{\arg\min} \sum cut(u^n) \tag{3}$$

where $cut(u) = cut(C) = \sum_{i,j \in C} weight(i, j)$, which corresponds to the sum of the weights for the edges that are cut (shown in red dashed line in figure 4). By solving the min-cut of the graph, we can obtain the region grown from seed 1, which are the pixels connected to the source after cutting.

### 1.2.2 Merging two cells and splitting a cell

Once we know how to add a cell, merging and splitting are straightforward. When merging two cells, we consider the regions with two old labels as the seed and do morphological dilation for the seed to get the foreground. The filter of morphological dilation is a sphere using "crop size" as the radius. Then we do the boundary refinement for the given seed and foreground. When splitting a cell, we delete the old label first and add two seeds simultaneously. Then we calculate the foreground and refine the boundary for each seed. As the other seed will be considered as the sink when we refine the current seed, the grown regions of the two seeds will not overlap with each other.

### 1.2.3 Smart click correction

The boundary-refine algorithm requires the user to consistently click at the center of the cell. Failure to do so may result in a suggestive boundary that encompasses only the seed region. This limitation arises from situations where a user clicks at the edge of a cell and all pixels surrounding the seed exhibit relatively high positive principal curvature since using the edges around the seed as the boundary in such cases leads to an even smaller $cut(u)$ compared to the correct boundary.

To address this issue, the "smart click correction" algorithm leverages the pixel within the seed as the source and identifies pixels with negative principal curvature as the sink. Using the same graph discussed earlier, the algorithm calculates the shortest path between the source and sink. When the "smart click correction" checkbox is activated, all pixels along the shortest path are used to generate a new seed, which will replace the original seed in the boundary refinement algorithm.

## 2 Label fusion detail

After aligning the labels assigned by various annotators with a designated ground truth, the subsequent step entails amalgamating these annotations to delineate the boundary of the ground truth. This process is underpinned by the principle of weighted voting, wherein the contribution of each annotator is influenced by factors such as the reviewer's expertise and the annotator's own experience. But there are some additional rules we applied to solve some overlapping issues. For two ground truth labels partially overlapping and two ground truth labels having different confidence scores, the ground truth label with the higher confidence score will keep the same. If more than 66% pixels of the lower confidence score ground truth are overlapping with the other label, it will be removed. If less than 33% of its pixels are overlapping with the other label, it will be kept but only use the pixels not overlapping with the other label. Otherwise, the non-overlapping pixels of lower confidence score ground truth will be considered undefined masks. For two ground truth labels partially overlapping and two ground truth labels having the same confidence score, we will sort annotators by their annotation experience, and treat the label created by a more experienced annotator as the one with a higher confidence score, then do the same thing we did to two ground truths with different confidence scores.

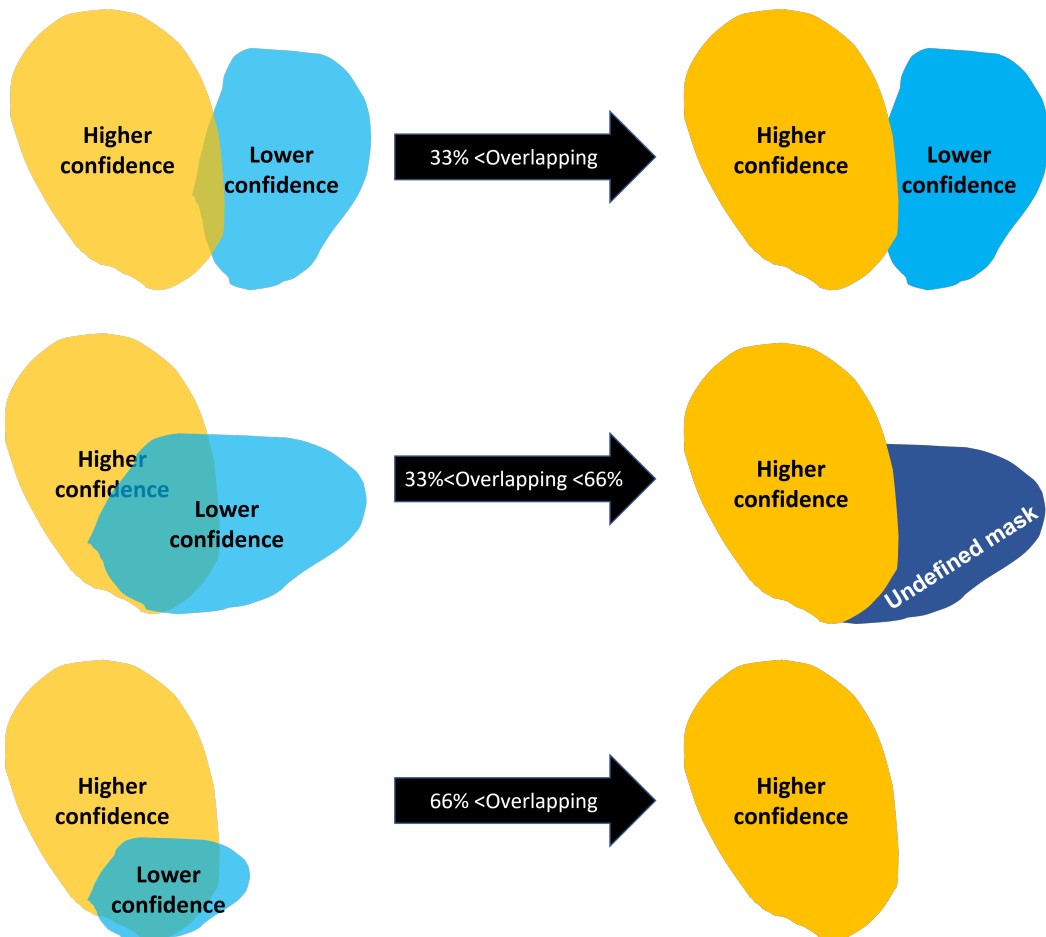

Figure 5: Example of the solution for overlapping problems

# 3 Experiment detail

## 3.1 Experiment setting

Quite a few methods set the minimum and maximum cell size/volume as hyperparameters. We report the average cell sizes (each from $10\%$ sample cells in the corresponding image) here so that readers can copy them directly when testing the existing methods. In Zebrafish 1, the cell size is around 2500. In Zebrafish 2, the cell size is around 400. In Drosophila 1, the cell size is around 4000. In Drosophila 2, the cell size is around 10000. In Mus Musculus 1, the cell size is around 2500. Other settings for each method are summarized below.

### 3.1.1 Cellpose

The developers of Cellpose have provided the pre-trained models to do instance segmentation for 3D images on their GitHub. We adjust the cell diameter and z-aspect for both "cyto" and "nuclei" models as shown in table 1. Everything else is the same as the default.

Table 1: Cellpose parameters

| Name | diameter | z-aspect |
|------|----------|----------|
| zebrafish 1 | 20 | 5.4 |
| zebrafish 2 | 5 | 1.0 |
| Drosophila 1 | 30 | 1.0 |
| Drosophila 2 | 35 | 1.0 |
| Mus Musculus | 30 | 1.0 |

### 3.1.2 QCAnet

The developers of QCAnet have provided the pre-trained models to do instance segmentation for 3D images on their GitHub. All settings remained as default.

### 3.1.3 StarDist

The developers of StarDist have provided the pre-trained models to do instance segmentation for 3D images on their GitHub. Moreover, they have provided a detailed tutorial on how to use their 2D segmentation model on GitHub. We referred to their tutorial and only changed the path of the model from 2D to 3D. Other settings remained as default.

### 3.1.4 3Dsuite

The developers have provided a Fiji plug-in for the 3d Suite method, which has included several different segmentation algorithms such as 3D watershed, 3D spot segmentation, 3D iterative thresholding, etc. We select the 3D iterative thresholding algorithm since it yields the best segmentation performance. The main hyperparameters that need to be tuned are the minimum cell volume, maximum cell volume (both in terms of the number of pixels). We adjust the minimum and maximum volumes according to the cell sizes in each image.

### 3.1.5 Vaa3D

The developers of Vaa3D have provided a software for 3D bioimage processing. In the Vaa3D software, the only segmentation algorithm is based on gradient vector flow. The hyperparameters that need to be tuned include: the iterations of diffusion, fusion threshold, and the minimum cell size. We employ the default setting for the first two hyperparameters while the third one is based on the cell sizes in each image.

## 3.2 Other methods we tried

There are many famous semantic segmentation methods, but we didn't compare them with instance segmentation methods. If the cells are far away from each other, we can use some post-processing to split them easily, but for data like NIS3D, they are not very useful. An example is shown in Figure 6.

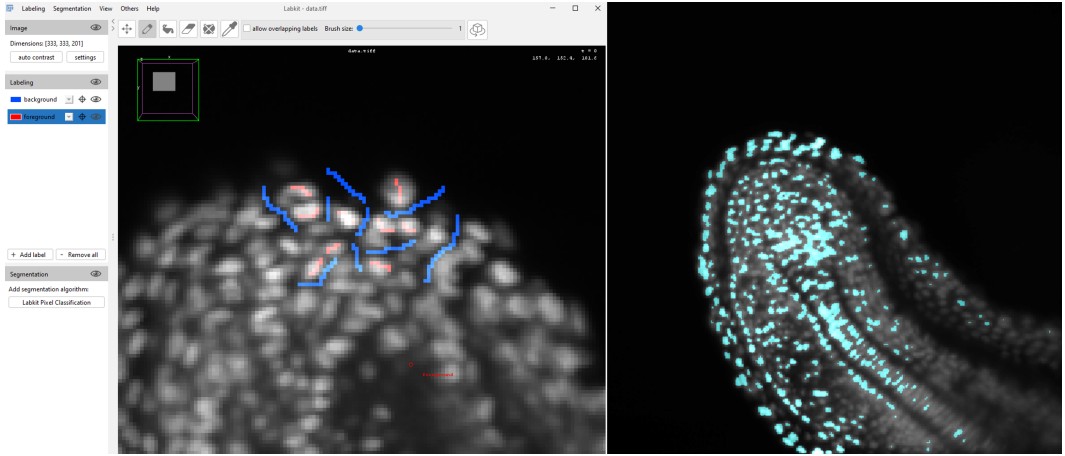

Figure 6: An example of semantic segmentation result (Labkits). The left is the human input, and the right is the segmentation result. The connected component can only give several instances as they are highly connected in 3D space.

We also tested the methods from MorphoLibJ, which provides a group of unsupervised 3D instance segmentation methods. We tried a wide range of parameters, but the result is much worse than the other method we tested and we did not put it into baseline methods. An example is shown in Figure 7.

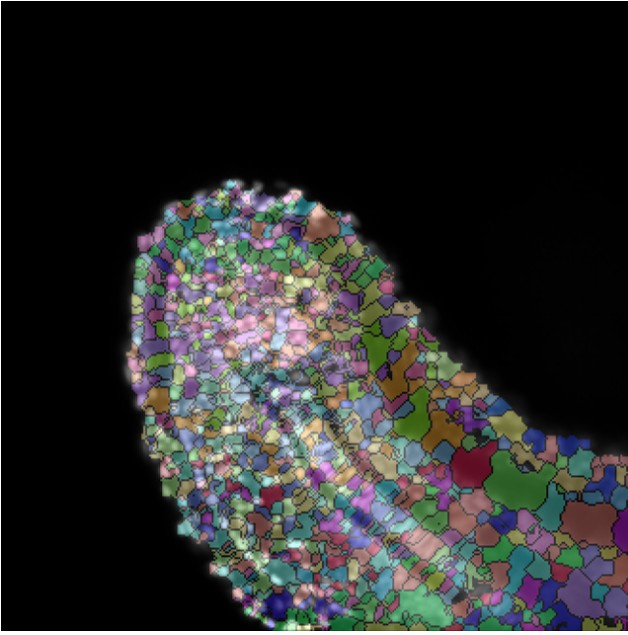

Figure 7: Segmentation result by morphological segmentation in MorphoLibJ. This result is based on parameters: "object image" and "gradient radius = 3"

# 4 Other questions

## 4.1 Compared with STABLE

An alternative approach involves combining the ground truth created solely by human annotators using the STAPLE algorithm with conventional unweighted F1 scores or Intersection over Union (IoU) scores. This particular method attributes uniform weight to all cells identified through the STAPLE approach, whereas cells exclusively identified by a single annotator are typically classified to a background. Besides this, STAPLE algorithm will still generate misleading ground truth when annotators give conflicted labels, rather than giving an undefined mask like our method. Drawing from our experience, we have observed a preference among users for the identification of strong cells, and a willingness to accept the omission of inconspicuous cells that might elude human perception. This differential weighting within our proposed method aligns with this user preference.

## 4.2 Why three annotators

The intricacies of annotating 3D nuclei are profound, and it is almost inevitable that within this complex process, discrepancies such as incomplete or erroneous labels may arise (check Figure Example of incomplete labels and noise label of C.elegans dataset). Without proofreading from different annotators, the quality of ground truth cannot be guaranteed. In contrast, NIS3D not only boasts a larger number of annotated instances, but also encompasses a wider array of species, experimental situations, and developmental stages. Besides, NIS3D carefully designed a strategy to fuse the ground truth from different annotators, which increase the boundary accuracy and significantly reduces the chance of incomplete labels.

**Example of incomplete labels and noise label of C_elegans_nuclei dataset**

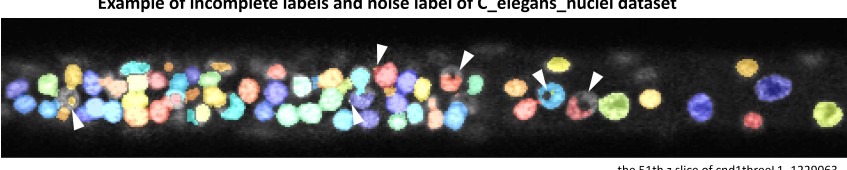

the 51th z slice of cnd1threeL1_1229063

Figure 8: Example of incomplete labels and noise labels

## 4.3 Supervised learning

We fine-tuned Stardist model to show the ability of the dataset to improve the existing model. Here we use in-image split (50% of the image as the training set and the other 50% as the test set for all images) and default train parameters provided by Stardist. We didn't use the confidence score and treated the undefined mask as pixels without cells. With the default setting, Stardist model shows significant improvement under the measure of both W-F1 score and W-SEG score.

**Stardist pre-trained model result**

| data | W-F1 | W-Precision | W-Recall | W-IoU | W-SEG |
| --- | --- | --- | --- | --- | --- |
| Zebrafish 1 | 0.586 | 0.770 | 0.473 | 0.263 | 0.192 |
| Zebrafish 2 | 0.529 | 0.977 | 0.363 | 0.380 | 0.183 |
| Drosophila 1 | 0.924 | 0.881 | 0.972 | 0.691 | 0.586 |
| Drosophila 2 | 0.684 | 0.526 | 0.979 | 0.563 | 0.325 |
| Mus Musculus | 0.594 | 0.789 | 0.476 | 0.256 | 0.191 |

**Stardist fine-tuned model results**

| data | W-F1 | W-Precision | W-Recall | W-IoU | W-SEG |
| --- | --- | --- | --- | --- | --- |
| Zebrafish 1 | 0.942 | 0.969 | 0.915 | 0.800 | 0.687 |
| Zebrafish 2 | 0.804 | 0.750 | 0.867 | 0.707 | 0.378 |
| Drosophila 1 | 0.971 | 0.995 | 0.947 | 0.860 | 0.772 |
| Drosophila 2 | 0.985 | 0.993 | 0.978 | 0.873 | 0.822 |
| Mus Musculus | 0.793 | 0.982 | 0.664 | 0.597 | 0.475 |

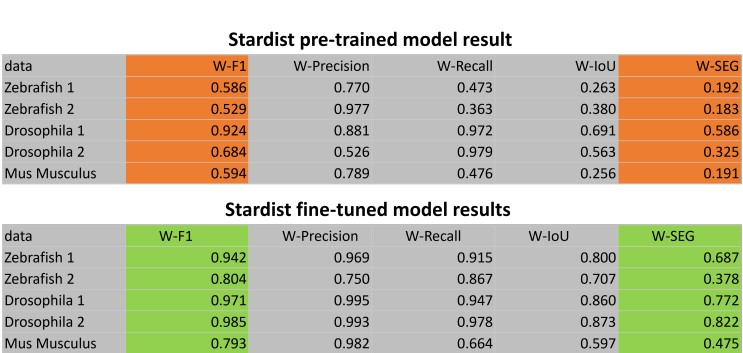

Figure 9: Stardist before and after a simple fine-tuning for in-image split

## 4.4 Evaluation metrics design

Choosing the correct metric that adequately reflects the biological nature is important but usually neglected. When employing the existing metric of the cell tracking challenge and the 2018 Data Science Bowl, three distinct issues encountered deviate result from human intuition, thereby affecting the evaluation process. Firstly, some methods can only detect the central portions of cells. Considering the true positive criteria in those existing benchmarks requiring 50% or more IoU score, even when a cell is successfully identified by 3D suite, its precision and recall may fall short of human expectations. This disparity becomes particularly relevant for biological studies that prioritize cell location, rendering the IoU score less informative. Secondly, approaches like Vaa3D yield favorable foreground detection outcomes but are plagued by a pronounced under-segmentation dilemma, existing metrics can't show their advantage for foreground detection. Thirdly, users express a preference for the detection of robust cells and can tolerate the omission of inconspicuous ones that may be overlooked by a human. To address these issues, we have reformulated the evaluation metric to align more closely with our specific objectives. For instance, a high W-F1 score coupled with a low W-SEG score now indicates successful cell detection while indicating room for boundary enhancement. Similarly, a high W-IoU score combined with a low W-SEG score signifies accurate foreground detection, while highlighting potential over-segmentation or under-segmentation concerns.