# OpenReview forum: "NIS3D: A Completely Annotated Benchmark for Dense 3D Nuclei Image Segmentation"
_NeurIPS.cc/2023/Track/Datasets_and_Benchmarks — NeurIPS 2023 Datasets and Benchmarks Poster_

### Official Review · Reviewer_W2ro · 2023-07-06
**A new dataset for dense 3D nuclei segmentation and benchmark for unsupervised learning**

**Rating:** 6
**Confidence:** 3
**Clarity:** Yes, this paper is well organized.

**Strengths:**

They collected their own data (Zebrafish 1) and made a public dataset.

This is the first benchmark that provides densely annotated cell in 3D volumes. Benchmarking 3D instance segmentation is more challenging than 2D mainly because constructing dataset is prohibitively laborious. Their proposed benchmark will be useful in developing 3D nuclei segmentation model in the future.

Recognizing the inherent ambiguity and potential biases in nuclei segmentation, the researchers introduced a confidence score as a means to establish a more reliable metric for their benchmark evaluation.

**Additional Feedback:**

No

**Correctness:**

The labels have been created by three skilled annotators, and their annotations have been intelligently fused to ensure accuracy using the confidence score. The evaluation metric for the dataset has been clearly and appropriately defined.

**Documentation:**

I could access the dataset and details of the dataset are explained.

**Limitations:**

Limitation is thoroughly discussed.

**Opportunities For Improvement:**

Supervised or Semi-supervised benchmarks seem to be more impactful, since developing an unsupervised model is challenging and sometimes unreliable, even if they present an unsupervised benchmark. Existing supervised methods such as 3D Mask R-CNN could be evaluated if they divide the images into 5 folds.

The dataset size, comprising only 5 images, is relatively smaller compared to other datasets, despite its high-volume nature. To enhance the credibility of the benchmark and enable a more comprehensive evaluation of existing supervised models, I strongly recommend increasing the dataset size to at least 10 images.

**Relation To Prior Work:**

Yes, they compared NIS3D with existing datasets and explained the difference in Table 1.

**Summary And Contributions:**

They constructed a dataset NIS3D for 3D embryonic cell image segmentation which contains annotations of 5 volumes (22000+ cells) from multiple species. NIS3D provides large-volume images with high cell densities, various SNR, shapes and brightness. Well trained annotators made instance segmentation labels and their labels were fused with the confident scores to represent an agreement between the annotators. Existing 3D cell segmentation methods showed unsatisfactory results on NIS3D.

---

> ### Author Response · Authors · 2023-08-21
>
> Thank you for your valuable suggestion!
>
> You can find the figures and tables mentioned in this comment here: https://drive.google.com/drive/folders/15Vle8Y_frB4auemW-_hunIa7f30DnAWJ?usp=sharing
>
> **About supervised learning:**
>
> 3D Mask R-CNN is generally not a popular model in this area. Despite the advances made by Mask R-CNN, it often underperforms due to the low signal-to-noise ratios and dense packing of nuclei in typical fluorescence microscopy datasets [1]. Instead, we fine-tuned Stardist model to show the ability of the dataset to improve the existing model. Here we use in-image split (50% of the image as the training set and the other 50% as the test set for all images) and default train parameters provided by Stardist. We didn’t use the confidence score and treated the undefined mask as pixels without cells. With the default setting, Stardist model shows significant improvement under the measure of both W-F1 score and W-SEG score (check Table Stardist pre-trained model result and Stardist fine-tuned model results for result).
>
> **About the dataset size:**
>
> Considering the cell number of each image, NIS3D has already provided a large number of annotated cells covering a wide array of species, experimental situations, and developmental stages. Adding more images to NIS3D is a good suggestion and currently, we’ve started work on it, NIS3D will have more images of more kinds of species over time. We are currently adding one more Mus Musculus into the NIS3D. It’s not easy and needs more time, but we promise we will finish it before the camera-ready version.
>
> [1] Martin Weigert, Uwe Schmidt, Robert Haase, Ko Sugawara, and Gene Myers. Star-convex polyhedra for 3d object detection and segmentation in microscopy. In Proceedings of the IEEE/CVF winter conference on applications of computer vision, pages 3666–3673, 2020.

---

### Official Review · Reviewer_d7Xp · 2023-07-19
**The paper presents a dataset which consists of segmented 3D cells for development and evaluation of 3D cell segmentation methods.**

**Rating:** 8
**Confidence:** 3
**Clarity:** The paper is clearly written.

**Strengths:**

- The dataset has both 3D data and 3D annotation labels.
- It is much larger than other similar datasets.
- The cell segmentations were carried out by three annotators. Variability among the annotators is computed as a confidence score and included in the dataset.
- The authors carry out an experimental comparison of several 3D segmentation methods, showing the utility of the dataset.

**Additional Feedback:**

.

**Correctness:**

The claims made in the submission are correct. The dataset is created in technically sound way.

**Documentation:**

The paper is clearly written, and a user should be able to get enough information about the dataset from the paper.

**Ethics:**

I don't think there are any ethical concerns with this submission.

**Limitations:**

The authors discuss the limitations of the dataset. The main limitation is, although the dataset is larger than its counterparts, it still is relatively small and may not be an ideal candidate to both train and test methods.

**Opportunities For Improvement:**

The authors could also carry out experiments to evaluate how/whether the dataset can be used to refine pre-trained models. The authors state that the dataset is not large enough to be used to both train and test algorithms. However, it could be split into two subsets, where on subset could be used to refine algorithms trained on other datasets. This could help evaluate if small to moderate size datasets could be used to improve performance of pre-trained models.

**Relation To Prior Work:**

It references prior work, and compares the dataset with other existing 3D datasets.

**Summary And Contributions:**

This work presents a dataset that consists of manually segmented 3D cells to be used in development and assessment of 3D cell segmentation algorithms.

It is an interesting work. 3D nucleus segmentation is a challenging problem. This dataset provides a valuable resource to support algorithm development and evaluation of algorithm performance. The dataset was annotated by three annotators. In addition to nucleus segmentations, the dataset contains a confidence score, as a measure of variability across three annotators, for each segmented nucleus. This could be very useful in comparing an algorithm's performance against inter-annotator variability. The authors also use the dataset in experimental review of multiple 3D segmentation methods.

---

> ### Author Response · Authors · 2023-08-21
>
> Thank you for your positive comments and suggestions.
>
> **Default train/test split settings:**
>
> We will provide two suggestive training/test split settings, one is an in-image split setting and one is a cross-image split setting. For the in-image split, we use 50% of the image as the training set and the other 50% as the test set. For cross-image split, we use zebrafish 2, Drosophila 1, and Mus Musculus as the training set and the rest as the test set. We are currently adding one more Mus Musculus into the NIS3D. It’s not easy and needs more time, but we promise we will finish it before the camera-ready version. The new Mus Musculus will be added to the test set of the cross-image split setting.

---

### Official Review · Reviewer_Rsqm · 2023-07-21
**A useful 3d nuclei segmentation dataset, yet with limitations regarding its use as benchmark**

**Rating:** 6
**Confidence:** 5
**Clarity:** The paper is well written and easy to…

**Strengths:**

Fully annotated 3d nuclei data is a scarce resource; To this end, the novel data describes in this work constitutes a very valuable contribution.
The proposed evaluation metric, which takes into account human annotator agreement, is very interesting.

**Additional Feedback:**

In summary, this is a very valuable dataset. To make it of direct use to the method development community, however, a train/test split has to be defined, ideally avoiding in-image splitting. Furthermore, existing densely annotated 3d nuclei data as employed in the StarDist3d work needs to be discussed, as well as related methodological work on incorporating rater confidence into evaluation metrics.

In the Discussion Phase, the authors have addressed the above points convincingly: They have shown annotation inaccuracies in existing 3d nuclei data, provided train/test splits, and elaborated on the relation to other approaches that deal with multiple annotations.

**Correctness:**

The precise train/test split of the data as used for benchmarking should be clearly defined in the paper to allow for future direct comparability of other approaches.

**Documentation:**

Documentation appears sufficient

**Ethics:**

No concerns

**Limitations:**

The authors have clearly laid out a limitation of their data, namely that for some model organisms, their data only comprised a single image.

**Opportunities For Improvement:**

An existing 3d nuclei dataset that has been employed for benchmarking in methodological work on instance segmentation is not mentioned:
"3D nuclei instance segmentation dataset of fluorescence microscopy volumes of C. elegans" available here:
https://zenodo.org/record/5942575#.YqD3vpDMKrM
It contains 28 3d images with >500 nuclei each, yielding a total of ca 15.000 densely annotated nuclei (albeit only one and not three annotations)
It also comes with a definition of a specific train/val/test split.
This data and split has, among others, been used in the StarDist3d work (Weigert et al. WACV 2020).
Please include this data in your discussion of existing benchmarks.

The proposed evaluation metric is very interesting, yet a discussion of respective related work is missing. Please put the metric into the comprehensive context recently provided by the Metrics Reloaded consortium (https://doi.org/10.48550/arXiv.2206.01653). Furthermore, please discuss related work on incorporating annotator agreement (or confidence) into evaluation metrics -- this is currently not discussed at all.

The presented benchmarking appears to be limited to off-the-shelf methods that are either non-learnt or pre-trained on other (diverse) data.  While the authors give reasons for this, it still renders the supervised methods not directly comparable.

To make the data useable for future benchmarking of novel instance segmentation methodology devised in the computer vision community, a default train/test split should be prominently defined in the paper. Furthermore, the core limitation of comprising only a single image for three of the model organisms  (as mentioned by the authors, see below), in particular the need for in-image splits mentioned by the authors, would severely limit an assessment of the generalization performance of segmentation methods. Cross-organism splits could also be an option -- please discuss, as this would alleviate the above issue.

**Relation To Prior Work:**

see "opportunities for improvement"

**Summary And Contributions:**

The authors contribute a set of five 3d fluorescence microscopy images of cell nuclei in developing model organisms with full ground truth instance segmentations by three human annotators. Furthermore, they propose an evaluation metric that takes into account annotator consensus, and benchmark popular off-the-shelf 3d nuclei segmentation methods on their data.

---

> ### Author Response · Authors · 2023-08-21
>
> Thank you for your valuable suggestion!
>
> You can find the figures and tables mentioned in this comment here: https://drive.google.com/drive/folders/15Vle8Y_frB4auemW-_hunIa7f30DnAWJ?usp=sharing
>
> **About the C_elegans_nuclei dataset:**
>
> Another existing dataset, C_elegans_nuclei has been used for model development but suffers from two major limitations. Firstly, it contains only one species of animal without enough diversity. And secondly, the annotations are from only one annotator. The intricacies of annotating 3D nuclei are profound, and it is almost inevitable that within this complex process, discrepancies such as incomplete or erroneous labels may arise (check Figure Example of incomplete labels and noise label of C_elegans_nuclei dataset). Without proofreading from different annotators, the quality of ground truth cannot be guaranteed. In contrast, NIS3D not only boasts a larger number of annotated instances, but also encompasses a wider array of species, experimental situations, and developmental stages. Besides, NIS3D carefully designed a strategy to fuse the ground truth from different annotators, which increase the boundary accuracy and significantly reduces the chance of incomplete labels.
>
> We will add C_elegans_nuclei to Table 1 of the main paper and cite the related works.
>
> **Related work of evaluation metric:**
>
> We checked the popular benchmarks within this field but didn’t find other work incorporating annotator agreement into evaluation metrics. Please let us know if we ignored any.
>
> As also mentioned by Metrics Reloaded consortium, choosing the correct metric that adequately reflects the biological nature is important but usually neglected [1]. Initially, our intention was to employ the proposed metric of the cell tracking challenge and the 2018 Data Science Bowl. Yet, during our investigation, we encountered three distinct issues that deviate from human intuition, thereby affecting the evaluation process.   Firstly, some methods like 3D suite iterative thresholding can only detect the central portions of cells. Considering the true positives criteria in those existing benchmarks requiring 50% or more IoU score, even when a cell is successfully identified by 3D suite, its precision and recall may fall short of human expectations. This disparity becomes particularly relevant for biological studies that prioritize cell location, rendering the IoU score less informative. Secondly, approaches like Vaa3D yield favorable foreground detection outcomes but are plagued by a pronounced under-segmentation dilemma, existing metrics can’t show their advantage for foreground detection. Thirdly, users express a preference for the detection of robust cells and can tolerate the omission of inconspicuous ones that may be overlooked by a human. To address these issues, we have reformulated the evaluation metric to align more closely with our specific objectives. For instance, a high W-F1 score coupled with a low W-SEG score now indicates successful cell detection while indicating room for boundary enhancement. Similarly, a high W-IoU score combined with a low W-SEG score signifies accurate foreground detection, while highlighting potential over-segmentation or under-segmentation concerns.
>
> We will cite the Metrics Reloaded in our paper.
>
> **Default train/test split settings:**
>
> We will provide two suggestive training/test split settings, one is an in-image split setting and one is a cross-image split setting. For the in-image split, we use 50% of the image as the training set and the other 50% as the test set. For cross-image split, we use zebrafish 2, Drosophila 1, and Mus Musculus as the training set and the rest as the test set. We are currently adding one more Mus Musculus into the NIS3D. It’s not easy and needs more time, but we promise we will finish it before the camera-ready version. The new Mus Musculus will be added to the test set of the cross-image split setting.
>
> [1] arXiv:2206.01653

---

> > ### Comment · Reviewer_Rsqm · 2023-08-28
> > **Thank you for your comprehensive reply**
> >
> > -- The Figure on annotation inaccuracies in the existing C. elegans dataset is very convincing regarding the distinct benefit of your work
> >
> > -- Regarding the evaluation metric, I was thinking of the following as potential related work:
> > Warfield et al., "Simultaneous Truth and Performance Level Estimation (STAPLE): An Algorithm for the Validation of Image Segmentation", IEEE Trans Med Imaging. 2004 Jul; 23(7): 903–921 https://doi.org/10.1109%2FTMI.2004.828354
> > It would be great if you could discuss the relation of this approach to your work.
> >
> > -- The suggested in-image and cross-image split settings (and resp. new data) will be highly beneficial for future benchmarking

---

> > > ### Author Response · Authors · 2023-08-29
> > >
> > > Thank you for your suggestions!
> > >
> > > **Regarding the Evaluation Metric:**
> > >
> > > Please let me know if my understanding of your question is inaccurate.
> > >
> > > The STAPLE algorithm holds a prominent position as an approach for fusing multiple annotations into a definitive ground truth and it can also evaluate the "sensitivity" and "specificity" of each annotator.
> > >
> > > There are two ways to use STAPLE algorithm to evaluate segmentation algorithms.
> > >
> > > Firstly, the segmentation algorithm’s result can be served as one of the input sources of the STAPLE algorithm. This enables the utilization of STAPLE's computed "sensitivity" and "specificity" metrics to characterize the algorithm's performance. However, this approach does present certain limitations. Notably, the algorithm's result serves a dual purpose, acting as both the foundation for generating the ground truth and the subject of evaluation. This dual utility could potentially introduce bias into the evaluation, particularly in scenarios with a limited number of human annotators. Additionally, the metrics provided, "sensitivity" and "specificity," while informative, might not offer a comprehensive representation of the algorithm's performance. Furthermore, the iterative nature of STAPLE algorithm can be time-consuming when dealing with data like nuclei segmentation, which have thousands of unordered cells.
> > >
> > > Alternatively, a second approach involves combining the ground truth created solely by human annotators using the STAPLE algorithm with conventional unweighted F1 scores or Intersection over Union (IoU) scores. This particular method attributes uniform weight to all cells identified through the STAPLE approach, whereas cells exclusively identified by a single annotator are typically classified to a background. Besides this, STAPLE algorithm will still generate misleading ground truth when annotators give conflicted labels, rather than giving an undefined mask like our method. Drawing from our experience, we have observed a preference among users for the identification of strong cells, and a willingness to accept the omission of inconspicuous cells that might elude human perception. This differential weighting within our proposed method aligns with this user preference.

---

> > > > ### Comment · Reviewer_Rsqm · 2023-08-30
> > > >
> > > > Thank you for elaborating on STAPLE. The "second approach" you mention is what I had in mind, and the relation you describe with your work is convincing. Please include this in your revised manuscript.
> > > >
> > > > My main concerns have been addressed, and I am raising my score accordingly.

---

### Official Review · Reviewer_bNXG · 2023-07-21
**Exhaustive 3D Cell Segmentation**

**Rating:** 8
**Confidence:** 3
**Clarity:** Yes the presentation is clear.

**Strengths:**

The paper is well-written and the ideas are well-presented. It is easy to read and understand.
Details on the dataset and the benchmark tests are included.

Authors provide exhaustive labels. They used multiple annotators to label the dataset and also presented how they merged the labels of the annotators. This also can be used as a guideline in future studies.

They used data from 4 different samples which increases the variation in the dataset and makes the dataset more generalizable.



**Additional Feedback:**

none

**Correctness:**

The reviewer thinks the dataset is constructed in a sound way.


**Documentation:**

Yes. the dataset is well explained. the benchmark can be released as codes if possible. the dataset is made available on google drive but I would suggest the authors to release the final version at a more dependable (permanent) repository and also get a DOI.

**Ethics:**

no ethical concerns

**Limitations:**


There are few claims/statements in the paper that are neither backed up with data/findings nor by proper references to the literature
i) Authors state that (line 74-75), using the presented tool speeds up the annotation workflow. However the did not present any data in this direction. In some cases, cleaning the labels generated by an automated model can be more tedious and time-consuming than doing the labelling from scratch. Did the authors at least test on a few images how speed-up benefit is achieved?
ii) Line 145: Princut method can automatically generate boundaries for user-identified cells, and the method is sensitive to weak intensity changes, which can reduce the human bias in low-quality regions.  Is this claim just observational or do the authors have quantitative analysis data to support this claim?



**Opportunities For Improvement:**

The major limitation of the study is about 3D labeling and evaluations. The current version of the work reads like all the labeling is done on 2D slices, and the volumetric correlation between the slices is not used. However, one important aspect of 3D labeling is that the cell surfaces in 3D should be continuous (and smooth?). However, the current approach of labeling each slice in 2D does not guarantee this. It is not clear if the authors take any necessary steps to ensure the continuity of the cell borders in 3D


**Relation To Prior Work:**

Yes, the authors discussed how this work differs from the previous contributions

**Summary And Contributions:**

The authors present a volumetric image dataset, which is exhaustively labeled in 3D for cells. They also present a baseline benchmark on the dataset.

---

> ### Author Response · Authors · 2023-08-21
>
> Thank you for your positive comments and suggestions.
>
> You can find the figures and tables mentioned in this comment here: https://drive.google.com/drive/folders/15Vle8Y_frB4auemW-_hunIa7f30DnAWJ?usp=sharing
>
> **About the smoothness of the 3D boundary:**
>
> Yes, the 3D boundary of a cell should be continuous and smooth, and we achieve this from two aspects. Firstly, the semi-automatic annotation tool we designed, PrinCut, takes both morphological and intensity information of cells into consideration and uses the max-flow min-cut algorithm to give the suggestive 3D boundary for the user-identified cell. In this way, this algorithm can ensure the suggestive boundary is continuous and smooth in 3D. Secondly, we train the annotators to make sure the annotators understand the morphological property of the cells. Even though annotators draw the cell with brushes, they need to carefully look at different z-slices to make sure the boundary is reasonable.
>
> **About the speed:**
>
> Based on our experience, PrinCut makes our work on average 10 times more efficient compared with purely by hand drawing. We try to annotate the cell without PrinCut initially. But we find that annotators use about 30 seconds to draw the boundary on each z slice with a computer mouse. If a cell is shown in 10 different z slices, it will take about 5 minutes to simply draw the boundary. And annotators also need time to look back and forth at the data to determine the boundary, ensure the smoothness of the 3D boundary, and sometimes adjust the existing boundary for old annotation, which will result in average 7 minutes for each 3D boundary. It is also very difficult to let annotators maintain such efficiency all day. If we do it purely by hand, we estimate the total labor will be over 7000 hours.
>
> **About reduces human bias:**
>
> This is a claim based on observation. When we are annotating, we find that humans may ignore the weak gap between cells, especially when humans do not use the optimal brightness and contrast to look at the gap. But PrinCut can detect the weak intensity change if the change is continuous across the whole cell and give it as a suggestive boundary to allow humans to give a more careful investigation from different z slices (check Figure Example of the weak gap between cells as an example).

---

> > ### Comment · Reviewer_bNXG · 2023-08-25
> > **Thanks for the review replies**
> >
> > Thanks to the authors for replying to my comments.

---

### Official Review · Reviewer_x8hy · 2023-07-26
**A 3D annotation benchmark set**

**Rating:** 8
**Confidence:** 5
**Correctness:** Yes

**Strengths:**

This is by far the largest data set in this specific problem space (light microscopy 3D segmentation). A tremendous amount of thoughtful, careful annotation has been done here, which will be important in this problem space for benchmarking methods.

**Additional Feedback:**

Very cool, a ton of work!

**Clarity:**

The paper is clear in most places, but is rather colloquial for this style of paper (which isn't bad per se, it's just diff erent)

**Documentation:**

No hosting or maintenance plan is provided

**Limitations:**

Yes

**Opportunities For Improvement:**

- Other than by the distribution of the confi dence scores, there is a missed opportunity here to provide (or at least report on) inter-annotator variability at a large scale. The authors might consider providing individual annotator data
- I found the arrows in Figure 4 very confusing - if I understand the relationship between the arrow diagrams and the circle diagrams correctly, in the second and third case, it's not so much that some annotators provided "unrelated" annotations, it's that they did not provide any in that area, at least that overlap by 0.5 IOU, yes? That should be made more clear ("missing or unrelated", perhaps)
- I didn't understand this how ties between annotations of equal confi dence are resolved (supplemental line 117-119), I think this needs an edit
- It would be nice to have some quantifi cation of the amount of "unconfi dent" area in each embryo
- See comment below re: lack of connection to the broader problem space of instance segmentation

**Relation To Prior Work:**

This refers extensively to other work in the light microscopy analysis space, but not to other instance segmentation data sets (such as in electron microscopy or natural images). Some comparison should be added, in terms of size as well as comparison with how ie label assignment has been done in some of these cases (it's not wrong to do it differently if the difference is justified, but it would be good to compare and contrast).

**Summary And Contributions:**

The authors here present a densely annotated 3D instance segmentation benchmark set, containing 22000 nuclei from 5 images of  embryos of 3 species.

---

> ### Author Response · Authors · 2023-08-21
>
> Thank you for your positive comments and suggestions.
>
> You can find the figures and tables mentioned in this comment here: https://drive.google.com/drive/folders/15Vle8Y_frB4auemW-_hunIa7f30DnAWJ?usp=sharing
>
> **Figure 4:**
>
> Yes, your understanding is correct, the red circle in Figure 4 indicates the close label created by the annotator is “unrelated to the ground truth”. Use the 2/3 confidence score as an example (check Figure GT with 2/3 confidence score), annotator A and annotator B have created an annotation for this ground truth while annotator C didn’t create any annotation belonging to this ground truth, and the confidence score of this ground truth is 2/3.
>
> We will add this example to the paper.
>
> **Supplemental line 117-119:**
>
> We show the way to solve the overlapping issue for two ground truths with different confidence scores (in supplement lines 107-117), but we also need to solve the overlapping issue for two 1/3 confidence score ground truths. In this case, we will sort annotators by their annotation experience, and treat the label created by a more experienced annotator as the one with a higher confidence score, then do the same thing we did to two ground truths with different confidence scores.
>
> **Undefined mask area:**
>
> Here is the undefined mask area by pixel number and by ratio (check Table Undefined mask area). The mask also contains pixels belonging to the gap between blurry cells, which may be misleading.
>
> **Compared with EM data and natural data:**
>
> Let me know if we didn’t understand your question correctly.
>
> Compared with EM data and natural data, florescent images generally have a lower signal-noise ratio and smaller instance size (by pixel number). As a result, the cell boundaries created by different annotators usually present more variation. Besides this, when annotating 3D data, it is almost inevitable that a single annotator may give incomplete labels, as 3D data are hard to visualize directly. Considering these two reasons, the strategy of fusing annotation from different annotators can be more important. A good fusing strategy can not only improve the boundary accuracy but also significantly reduce the incomplete label number.
>
> In addition, natural images are generally 2D and EM data generally only focus on a very small 3D region due to their cost.

---

> > ### Comment · Reviewer_x8hy · 2023-08-30
> >
> > I thank the authors for addressing my comments and have raised my score accordingly - I look forward to seeing the revised manuscript soon!

---

### Decision · Program_Chairs · 2023-09-22

**Decision:**

Accept (Poster)

**Comment:**

The reviewers all liked the paper. The authors' response clarified most points raised by the reviewers. In view of that, the authors are strongly invited to take the feedback on board for the final version.